Factor structure of the happiness-increasing strategies scales (H-ISS): activities and coping strategies in relation to positive and negative affect

Al Nima Ali 1 alinor_1979@yahoo.co.uk
Garcia Danilo 1 2 danilo.garcia@neuro.gu.se danilo.garcia@icloud.com
1 Network for Empowerment and Well-Being, University of Gothenburg , Gothenburg , Sweden
2 Institute of Neuroscience and Physiology, Center for Ethics, Law and Mental Health (CELAM), University of Gothenburg , Gothenburg , Sweden
Zohar Ada
Electronic publication date: 2015 Jul 2
Publication date: 2015
Volume: 3
Electronic Location ID: e1059
Received 2015 Jan 15; Accepted 2015 Jun 8
Copyright: © 2015 Al Nima and Garcia
Copyright year: 2015
Copyright holder: Al Nima and Garcia
License: This is an open access article distributed under the terms of the Creative Commons Attribution License, which permits unrestricted use, distribution, reproduction and adaptation in any medium and for any purpose provided that it is properly attributed. For attribution, the original author(s), title, publication source (PeerJ) and either DOI or URL of the article must be cited.
License URL: https://creativecommons.org/licenses/by/4.0/

Keywords: Happiness, Happiness-increasing strategies, Negative affect, Well-being, Positive affect

Funding: AFA Insurance 130345 This study was supported by AFA Insurance (Dnr. 130345). The funders had no role in study design, data collection and analysis, decision to publish, or preparation of the manuscript.

==============================
Background. Previous research (Tkach & Lyubomirsky, 2006) shows that there are eight general happiness-increasing strategies: social affiliation, partying, mental control, goal pursuit, passive leisure, active leisure, religion, and direct attempts. The present study investigates the factor structure of the happiness-increasing strategies scales (H-ISS) and their relationship to positive and negative affect.

Method. The present study used participants’ (N = 1,050 and age mean = 34.21 sd = 12.73) responses to the H-ISS in structural equation modeling analyses. Affect was measured using the Positive Affect Negative Affect Schedule.

Results. After small modifications we obtained a good model that contains the original eight factors/scales. Moreover, we found that women tend to use social affiliation, mental control, passive leisure, religion, and direct attempts more than men, while men preferred to engage in partying and clubbing more than women. The H-ISS explained significantly the variance of positive affect (R2 = .41) and the variance of negative affect (R2 = .27).

Conclusions. Our study is an addition to previous research showing that the factor structure of the happiness-increasing strategies is valid and reliable. However, due to the model fitting issues that arise in the present study, we give some suggestions for improving the instrument.

Introduction

Happiness is of main interest for psychologists who study the strengths and the positive aspects of humanity. Researchers suggest that “people all over the world most want to be happy by achieving the things they value” (Diener, Oishi & Lucas, 2003, p. 420). Moreover, happiness has a protective role and a strong influence in our life against negative feelings and even mental health disorders such as depression (e.g., Joseph et al., 2004; Russell & Feldman Barrett, 1999; Watson et al., 1999). Although the definition of happiness also encompasses other measures of well-being, such as life satisfaction and harmony in life (Diener, 1984, Cloninger, 2004; Kjell et al., 2015), for this article we have focused on the affective component of happiness. Herein, the concept of happiness involves the absence of displeasure and negative affect and simultaneously the presence of pleasant emotions and positive affect (Joseph et al., 2004). In this context, the observed variance in happiness among individuals might partially depend on the way, or more specifically the strategies, people intentionally use to increase or maintain their happiness (Määttä, Stattin & Nurmi, 2002). Indeed, according to Sheldon & Lyubomirsky (2006) individuals focus their energy and behavior in a variety of different ways to achieve happiness, that is, intentional happiness-increasing strategies that lead to a diverse set of experiences. In this line of thinking, Tkach & Lyubomirsky (2006) developed a measure comprising eight different strategies. To the best of our knowledge, however, the factor structure of this scale has only been confirmed once among undergraduate American students (Tkach & Lyubomirsky, 2006) and once in a small sample of Swedish adolescents (Nima, Archer & Garcia, 2013). What’s more, in the Swedish adolescent sample, researchers found one of the clusters to comprise a set of activities that were avoided or less practiced in order to achieve happiness (e.g., ‘draw’, ‘read a book’, ‘studied’). These prevented activities scale was not part of the original eight scales found by Tkach & Lyubomirsky (2006). These two studies were also inconsistent and showed mixed results with respect to gender differences. For example, both studies showed that females reported using communal strategies (i.e., Social Affiliation) more often than males, but while American males reported engaging more frequently in strategies related to agency (i.e., Active Leisure) and suppression (i.e., Mental Control) compared to American females (Tkach & Lyubomirsky, 2006), Swedish females scored higher than Swedish males in all of these strategies (Nima, Archer & Garcia, 2013). These mixed and inconsistent results might mirror cultural and age differences between the American and Swedish sample. Here we use a relatively large sample of US-residents to investigate the factor structure of the original scales and to investigate gender differences.

Happiness-increasing strategies

In their original study, Tkach & Lyubomirsky (2006) identified, using at first an open-ended survey and then a factor analysis, eight clusters of happiness-increasing strategies:

1. Social Affiliation, which can be considered as social activities such as supporting and encouraging friends (i.e., communion).

2. Partying and Clubbing, which refers to celebratory behavior such as drinking alcohol.

3. Mental Control, a strategy composed of ambivalent intentional efforts aimed, on the one hand, at avoidance of negative thoughts and feelings and, on the other hand, proneness towards contemplation of negative aspects of life.

4. Instrumental Goal Pursuit, which indicates the person’s desire and attempt to change one’s self or situation by, for example, achieving academic goals (i.e., agency).

5. Passive Leisure, a strategy that refers to idleness and involves passive activities (e.g., “Surf the Internet”).

6. Active Leisure, a strategy depicting the tendency to engage in activities such as exercise and hobbies in order to maintain or increase happiness and at the same time to lower and cope with stressful experiences (i.e., agency).

7. Religion, which involves the performance of religious activities such as seeking support from faith (i.e., spirituality).

8. Direct Attempts, a proactive strategy aiming to cultivate a happy mood by, for example, smiling (i.e., agency).

Tkach & Lyubomirsky (2006) found that the strongest unique predictors of happiness were Mental Control, Direct Attempts, Social Affiliation, Religion, Partying and Clubbing, and Active Leisure. The strategy that was the most robust predictor of happiness, Mental Control, is defined as ambivalent intentional efforts aimed, on the one hand, at avoidance of negative thoughts and feelings and, on the other hand, proneness towards contemplation of negative aspects of life. In their study, the happiness-increasing strategies scales accounted for 52%, while the Big Five personality traits for 46% of the variance in happiness. Moreover, even after controlling for the contribution of personality, the strategies accounted for 16% of the variance in happiness. However, the strengths of the relationships between the strategies and happiness varied to a great extent. Tkach & Lyubomirsky (2006) also found that men prefer to be engaged in Active Leisure and Mental Control, whereas women report using Social Affiliation, Goal Pursuit, Passive Leisure, and Religion more frequently than men do. Moreover, their results indicate that Social Affiliation, Mental Control, and Direct Attempts partially mediate many effects of individuals’ personality upon happiness levels. The relationship between happiness and Openness was, however, completely mediated by Social Affiliation.

As earlier stated, to the best of our knowledge only a few studies have used the happiness-increasing strategies scales (H-ISS) in relation to measures of happiness. In Schütz and colleagues study (2013), high levels of positive affect and low levels of negative affect were associated to Social Affiliation (comprising communal or cooperation values), Instrumental Goal Pursuit (comprising agentic or autonomous, self-directed values), Active Leisure (also comprising agentic values), and spiritual strategies such as frequently seeking support from faith, performing religious activities, praying, and drinking less alcohol (i.e., the Religion happiness-increasing strategy). Among Swedish adolescents, these three strategies (Social Affiliation, Instrumental Goal Pursuit, and Active Leisure) are positively related to subjective well-being (Nima, Archer & Garcia, 2013). Nevertheless, none of these studies has investigated the original factor structure of the H-ISS. The first aim of the present study is to investigate the factor structure of the H-ISS constructed by Tkach & Lyubomirsky (2006). Our second aim is to investigate gender differences in the use of the strategies. Our third and final aim is to investigate the strategies’ relationship to the experience of frequent positive affect and infrequent negative affect.

Method

Ethical statement

After consulting with the Network for Empowerment and Well-Being’s Review Board we arrived at the conclusion that the design of the present study (e.g., all participants’ data were anonymous and will not be used for commercial or other non-scientific purposes) required only informed consent from the participants.

Participants and procedure

The participants (N = 1,050, age mean = 34.21 sd. = 12.73, 430 males and 620 females) were recruited through Amazon’s Mechanical Turk (MTurk; https://www.mturk.com/mturk/welcome). MTurk allows data collectors to recruit participants (called workers) online for completing different tasks in change for wages. This method for data collection online is an empirical tested valid tool for conducting research in the social sciences (see Buhrmester, Kwang & Gosling, 2011). Participants were recruited by the following criteria: US-resident and English fluency and literacy. Participants were paid a wage of 50 cents (American dollars) for completing the task and informed that the study was confidential and voluntary. The participants were presented with a battery of self-reports comprising the affect and happiness measures, as well as questions pertaining age and gender.

Instruments

Happiness-increasing strategies scales (H-ISS; Tkach & Lyubomirsky, 2006)

The instrument consists of 35 strategies/items which participants are asked to rate how often they use in order to increase their happiness (1 = never, 7 = all the time). The responses are originally organized in eight clusters or scales: Social Affiliation (e.g., “Support and encourage friends”), Partying and Clubbing (e.g., “Drink alcohol”), Mental Control (e.g., “Try not to think about being unhappy”), Instrumental Goal Pursuit (e.g., “Study”), Passive Leisure (e.g., “Surf the internet”), Active Leisure (e.g. “Exercise”), Religion (e.g., “Seek support from faith”), and Direct Attempts (e.g., “Act happy/smile, etc”.).

The positive affect and negative affect schedule (Watson, Clark & Tellegen, 1988)

Participants are instructed to rate to what extent they generally have experienced 20 (10 positive and 10 negative) different feelings or emotions during the last weeks, using a 5-point Likert scale (1 = very slightly, 5 = extremely). The 10-item positive affect scale includes adjectives such as strong, proud, and interested (Cronbach’s α = .90 in the present study). The 10-item negative affect scale includes adjectives such as afraid, ashamed and nervous (Cronbach’s α = .88 in the present study).

Results

Factor structure of the happiness-increasing strategies scales (H-ISS)

We conducted a confirmatory analysis using structural equation modeling (SEM) on the 35 items to estimate the eight factors of the H-ISS as suggested in the original article by Tkach & Lyubomirsky (2006). Tkach & Lyubomirsky’s original work (2006) showed that there are significant correlations between these eight scales, which potentially introduce the problem of multicollinearity. Thus, the data was analysed using SEM in order to control multicollinearity among the eight latent factors. The possible covariance between the eight different factors was taken into account in a hypothesized structural equation model (see Fig. 1). The analysis showed that chi-square value was significant (Chi2 = 2366.10, df = 531, p < .001), the goodness of fit index was .88 and the root mean square error of approximation for the default model was .06. Thus, while the fit index indicates that the model is not a good-fitting model, the root mean square error of approximation indicates a good-fitting model. Some modifications were performed in an attempt to develop a better fitting.

Figure 1 Hypothesized structural equation model of the eight latent factors of the happiness-increasing strategies scales according to Tkach & Lyubomirsky (2006).

After modifications (i.e., covariance between errors), to obtain a better fitting, two items were removed from the analyses (i.e., “Help others” and “Study”). Firstly, the item “Help others” had lower loading (.54) than the item “Support and encourage friends” (.61) and both have almost the same meaning. Also in this line, the item “Study” had lower loading (.41) than the item “Try to do well academically/raise grades” (.51) and both have the same meaning. The covariance between errors on the items “Go out to movies with friends” and “Stay home and enjoy quiet times” was −.28 . By adding correlated residuals, we assumed a correlation between these two items (i.e., “Go out to movies with friends” and “Stay home and enjoy quiet times”); a correlation that is not predicted by the latent variable, in this case passive leisure. We added these correlated residuals to improve the fit of the model. In short, the addition of the negative correlated residual of these items means that there is negative correlation between the items that is explained outside of our model.

After these further modifications, the chi-square value was still significant for the default model (Chi2 = 1885.39, df = 465, p < .001), the goodness of fit index for the default model was .90 and the root mean square error of approximation was .05. However, the chi-square statistic is heavily influenced by sample size (Kline, 2010), with larger samples leading to larger value and therefore, a larger likelihood of being significant. The goodness of fit index with a cut off value of .90 generally indicates acceptable model fit and the root mean square error of approximation with a cut off value of .05 also indicates good model fit (MacCallum, Browne & Sugawara, 1996). All the regression weights/loadings between scales and their items were significant at p < .001 (ranging from −.14 to .93) with the exception of the item “Try not to think about being unhappy”, which was significant at p = .003 and had a low loading (.11) within the mental control scale (see Fig. 2).

Figure 2 Structural equation model showing the correlations and standardized parameter estimates among the eight latent factors and among the latent factors and the items in the H-ISS.

Chi-square = 1885.39, df = 465, p = .001; goodness of fit index = .90 and the root mean square error of approximation = .05 (N = 1,050).

Reliability and correlations

The factors derived by the analysis showed good reliabilities (Cronbach’s alphas between .70 to .77) for five of the happiness-increasing strategies scales: social interaction, partying and clubbing, instrumental goal pursuit, religion and direct attempts. For the rest of the strategies (i.e., mental control, passive leisure, and active leisure) Cronbach’s alphas ranged from .52 to .65. Most of the inter-correlations between the happiness-increasing strategies were significant, ranging from −.08, p < .01 to .57, p < .01. The largest correlation being that between social affiliation and direct attempts (r = .57, p < .01) and the lowest correlation being that between social affiliation and mental control (r = − .08, p < .01) and between social affiliation and active leisure (r = − .08, p < .05). In other words, all significant correlations were significant at p < .01 with the exception of one correlation that was significant at p < .05 (see Table 1). Although some of the inter-correlations between the happiness-increasing strategies scales were not large, the results mirror those in the original article by Tkach & Lyubomirsky (2006), which ranged from −.09, p < .05 to .45, p < .001. In other words, suggesting that the eight strategies are related but different constructs.

Table 1 Correlations and reliability (Cronbach’s a in bold type in diagonal) coefficients, means and standard deviations for all happiness-increasing strategies scales (N = 1050).

Strategies	1	2	3	4	5	6	7	8	
Social Affiliation (1)	.77								
Partying Clubbing (2)	.30**	.75							
Mental Control (3)	−.08*	.06	.52						
Instrumental Goal Pursuit (4)	.45**	.26**	.01	.75					
Passive Leisure (5)	.24**	.25**	.24**	.22**	.52				
Active Leisure (6)	.34**	.26**	−.08*	.43**	.19**	.65			
Religion (7)	.11**	−.32**	−.01	.15**	−.04	.11**	.70		
Direct Attempts (8)	.57**	.21**	−.06	.45**	.24**	.33**	.19**	.76	
Mean	3.66	2.24	2.72	3.41	3.30	3.40	2.90	3.51	
Sd.	.75	.83	.76	.85	.59	.90	1.11	.92	
Notes.

* p < .05.

** p < .01.

Gender differences in the usage of the strategies

An independent samples t-test was used to investigate the differences between men and women in the happiness-increasing strategies. The results showed that women preferred using social affiliation, mental control, passive leisure, religion, and direct attempts more than men do to maintain or increase their happiness, while men preferred to engage in partying and clubbing more than women. See Table 2 for details.

Table 2 Means and standard deviations (±) for the usage frequency of strategy by men and women.

Happiness-increasing strategy	Men (N = 430)	Women (N = 620)	t	
Social interaction	3.59 ± .81	3.71 ± .70	−2.69**	
Partying and clubbing	2.32 ± .87	2.17 ± .79	2.95**	
Mental control	2.64 ± .77	2.77 ± .75	−2.63**	
Instrumental goal pursuit	3.36 ± .86	3.44 ± .85	−1.55	
Passive leisure	3.25 ± .61	3.34 ± .57	−2.34*	
Active leisure	3.44 ± .94	3.37 ± .87	1.28	
Religion	2.66 ± 1.03	3.06 ± 1.13	−5.85***	
Direct attempts	3.43 ± .98	3.57 ± .88	−2.38*	
Notes.

* p < 0.05.

** p < 0.01.

*** p < 0.001.

Happiness-increasing strategies scales and affect

The participants in this part of the study were 1,000 with an age mean = 34.22 (sd. = 12.73). The sample comprised 404 males and 596 females. That is, 50 participants were dropped from the original sample due to missing values on the affect measure. As demonstrated by Tkach and Tkach & Lyubomirsky’s original work (2006) and the confirmatory analysis conducted here, there are significant inter-correlations between these eight happiness-increasing strategies scales. Thus, the data was analysed using SEM in order to control multicollinearity among these eight latent factors. The results showed that the chi-square value was significant (Chi2 = 10.28, df = 1, p = .001). Again, the chi-square statistic is heavily influenced by sample size (Kline, 2010), with larger samples leading to a larger value and therefore, a larger likelihood of being significant. The goodness of fit index was 1.00, the incremental fit index was 1.00, and the Root Mean Square Error of Approximation fit statistic that was below 0. 10. All indicated that the model fit was acceptable (cf. Bollen, 1989; Browne & Cudeck, 1993).

The happiness-increasing strategies that predicted positive affect were: social affiliation (β = .16, p < .001), mental control (β = − .16, p < .001), instrumental goal pursuit (β = .20, p < .001), active leisure (β = .14, p < .001), religion (β = .07, p = .007) and direct attempts (β = .26, p < .001). Negative affect was predicted by social affiliation (β = − .14, p < .001), mental control (β = .35, p < .001), instrumental goal pursuit (β = .09, p = .01), passive leisure (β = .14, p < .001), active leisure (β = − .14, p < .001) and direct attempts (β = − .17, p < .001). The whole model showed a R2 = .41 for positive affect and .27 for negative affect (see Fig. 3). This means that 41% and 27% of the variance of positive affect and negative affect, respectively, were accounted for by the happiness-increasing strategies. In other words, 59% and 73% of variance of positive affect and negative affect respectively are predicted outside of our model.

Figure 3 Structural equation model of the eight happiness-increasing strategies scales and the correlations among the eight strategies and the paths from the strategies to positive and negative affect.

Chi-square = 10.28, df = 1, p = .001; goodness of fit index = 1.00, incremental fit index = 1.00; and the root mean square error of approximation = .096 (N = 1,000).

Discussion

The purpose of this article was to examine and provide a basis for understanding the factor structure of the happiness-increasing strategies found by Tkach & Lyubomirsky (2006) and also gender differences in these strategies. Moreover, the present study also assessed the relationships between the strategies and both positive and negative affect (cf. Tkach & Lyubomirsky, 2006).

We obtained a good fit of the overall model after some modifications—removal of two items and covariance between two errors. As expected, we found eight factors: Social Affiliation, Partying and Clubbing, Mental Control, Instrumental Goal Pursuit, Passive Leisure, Active Leisure, Religion and Direct Attempts (cf. Tkach & Lyubomirsky, 2006). These small differences between our modified model and the eight factor structure found by Tkach & Lyubomirsky (2006), may be explained or attributed to the differences between our sample and the sample in the original study with regard to, for example, ethnic backgrounds, age, and size sample. Our results also differ partially from the previous study by Nima, Archer & Garcia (2013) using a principal axis factoring procedure to extract the eight happiness-increasing scales (i.e., social interaction, mental control, partying, religion, self-directed, instrumental goal pursuit, active leisure, and prevented activities). For instance, some of the strategies (e.g., social interaction) found by Nima and colleagues (2013) were even differently labeled from the original scales by Tkach & Lyubomirsky (2006) and those found in the present study. In the Nima study (Nima, Archer & Garcia, 2013), the strategy labeled social interaction was a combination of items that were found by Tkach & Lyubomirsky (2006) and items “relating to how adolescents’ interact with others by pleasing them” (Nima, Archer & Garcia, 2013, p. 199). Related to this, as in Tkach and Lyubomirsky’s study, we found in the present study that the item “Savor the moment” loaded in the Social Affiliation factor. Nevertheless, although this, and other items for that matter, loaded accordingly to the original model proposed by Tkach and Lyubomirsky, it is plausible to question what it means that this item loads with the other behaviors involving social affiliation. After all, although both the quantity and quality of social affiliations influence people’s mental health, health behavior, physical health, and mortality risk (Umberson & Montez, 2010, p. 51), savoring the moment can definitely be practiced in isolation. In the original study, the items were generated in an open-ended survey among students, a population in which social affiliations are not only important but are also a recurrent part of their life as students attending a university. Perhaps this explains why the item loaded into the social affiliation factor from the very beginning. Our analysis, after all, was just confirmatory and therefore if exploratory analysis are conducted, the results may show that this specific item loads in another factor.

Descriptive analysis about difference between gender regarding happiness-increasing strategies indicated that both women and men report using social affiliation more frequently than other strategies. However, our findings show that women score higher in social affiliation, religion and passive leisure. Women, compared to men, preferred using social affiliation, religion and passive leisure more often to manage bad moods and to make themselves happy. Our result, combined with the result found by Tkach & Lyubomirsky (2006), confirms that women focus their attention on social relationships, religious activities and the inactive strategy of passive leisure more than men to increase and influence long-term happiness. In contrast to Tkach & Lyubomirsky (2006), our results show that women favor and tend to use mental control more than men to become happier. This is, however, in line with research suggesting that women try, more so than men, to avoid and suppress their negative thoughts and sad feelings as a happiness-increasing strategy (Nima, Archer & Garcia, 2013). Moreover, gender differences in the present study suggest that women try more than men to cultivate happy feelings and remove/avoid negative experiences using the strategy of direct attempts. Finally, in line with Nima, Archer & Garcia (2013), men, in contrast to women, use partying and clubbing more frequently. Men tend to use celebratory behavior such as drinking alcohol to lift away negative feelings and enjoy temporal happiness. According to these findings, women have greater tendency than men to handle and reduce negative mood, and to increase their happiness levels through social affiliation (e.g., receive support from friends), religion (e.g., seek support from faith), passive leisure (e.g., sleep), mental control (e.g., focus out negative aspects of life and ruminate), and direct attempts (e.g., act smile). Men used partying and clubbing (e.g., party, dance and go out to bars with friends) more than women did to control the consequences of their unhappiness and to try to improve their moods and to maintain their own continued happiness. All these differences mirror those in the literature suggesting that women, compared to men, report using social support more frequently (Thayer, Newman & McClain, 1994) and have a greater tendency to ruminate (i.e., mental control) about the causes and consequences of their unhappiness (Nolen-Hoeksema, 1991).

The results of the happiness-increasing strategies and affects model show that the happiness-increasing strategies significantly explain positive affect (R2 = .41) and negative affect (R2 = .27). This indicates that the variance in individual differences in happiness is related and linked with the happiness-increasing strategies. These findings are in line with Tkach & Lyubomirsky’s original study (2006) showing that the happiness-increasing strategies significantly explain .52 of the total variance in self-reported happiness, and also with findings among Swedish adolescents showing that the happiness-increasing strategies explain .43 of the variance in positive affect and .18 of the variance in negative affect (Nima, Archer & Garcia, 2013). The analysis indicates that social affiliation, instrumental goal pursuit, active leisure, religion and direct attempts predict and contribute uniquely to high levels of positive affect while mental control contributes to low levels of positive affect. This result confirms the original findings that underline the positive association between the happiness-increasing strategies (direct attempts, social affiliation, religion, partying and active leisure) and happiness while the mental control strategy is negatively associated with happiness (Tkach & Lyubomirsky, 2006). Moreover, our findings indicate also that negative affect is predicted negatively by social affiliation, active leisure and direct attempts strategies, while negative affect is positively predicted by mental control and passive leisure. In other words, certain types of behaviors of individuals such as interacting with friends, exercise and deciding to be happy may lead to decreases in negative affect, while certain behaviors such as focusing on negative aspects of life and sleep can lead to increases in negative affect. In general, these findings are in line with Tkach & Lyubomirsky’s study (2006). The result for the instrumental goal pursuit is unexpected because it associates positively with negative affect. However, instrumental goal pursuit was not a strong unique predictor (β = .09) of negative affect. The positive correlation between instrumental goal pursuit and passive leisure may lead to some construct overlap, which in turn makes instrumental goal pursuit a positive influence on negative affect. However, instrumental goal pursuit did not significantly contribute to happiness in the original study by Tkach & Lyubomirsky (2006).

Most happiness-increasing strategies predicted significantly positive affect and negative affect, ranging from β = .07, p < .001 to β = .35, p < .001, so these findings are nearly identical to the original study (Tkach & Lyubomirsky, 2006), ranging from β = .09, p < .05 to β = .48, p < .001. In other words, the variance of positive affect was accounted for largely enough by the happiness-increasing strategies. Although only 27% of the variance of negative affect was accounted for the strategies, most happiness-increasing strategies that predicted negative affect were significant at p < .001. Our findings suggest that about 73% of variance of negative affect is predicted outside of our model. This means that there are other variables (e.g., personality, demographic variables, and circumstantial factors) that may predict negative affect. Lyubomirsky and colleagues (2005) have indeed suggested three general categories of happiness predictors: (1) life circumstances and demographics, (2) traits and dispositions, and (3) intentional behaviors. The intentional behaviors are represented here as the happiness-increasing strategies. Thus, while intentional behavior may largely predict positive emotions, it seems like the other two happiness predictors (i.e., life circumstances and demographics and traits and dispositions) have a greater influence on negative emotions. This is also in line with a recent review of the literature on positive and negative affect suggesting a larger genetic component for negative affect than for positive affect (Cloninger & Garcia, 2015). In other words, positive affect can be enhanced more by what the individual makes of herself/himself intentionally than by her/his temperament traits; while negative affect is probably less influenced by these intentional activities due to negative affect’s genetic etiology.

Limitations and suggestions for future studies

One limitation is that the results are based on MTurk workers’ self-reports. Some aspects related to this data collection method might influence the validity of the results, such as, workers’ attention levels, cross-talk between participants, and the fact that participants get remuneration for their answers (Buhrmester, Kwang & Gosling, 2011). Nevertheless, a large quantity of studies show that data on psychological measures collected through Amazon’s Mechanical Turk meets academic standards and it is demographically diverse (Buhrmester, Kwang & Gosling, 2011; Paolacci, Chandler & Ipeirotis, 2010; Horton, Rand & Zeckhauser, 2011). Moreover, data on health measures collected through Amazon’s Mechanical Turk shows satisfactory internal as well as test-retest reliability (Shapiro, Chandler & Mueller, 2013). In addition, the amount of payment does not seem to affect data quality; remuneration is usually small, and workers report being intrinsically motivated (e.g., participate for enjoyment) (Buhrmester, Kwang & Gosling, 2011).

Moreover, some factors had low alphas, two items were removed, one path, not hypothesized, was added, and one item loaded in two factors; hence it is important that further research provides some evidence by cross-validating the model presented here and the one proposed by Tkach & Lyubomirsky (2006). For instance, as suggested by Tabachnick & Fidell, (2007, p. 760): “Extreme caution should be used when adding paths as they are generally post hoc and therefore potentially capitalizing on chance”. Nevertheless, also as suggested by Tabachnick & Fidell (2007), we used very conservative p levels (i.e., p < .001) as the criterion for adding post hoc parameters to the model. Finally, we had a large sample size that can easily lead to small standard errors, which is more likely to produce statistically significant results (here in our study even covariances among residuals/errors are significant). These covariances among errors are added on hypothesized structural equation model in order to improve the fit of the model. In sum, the model fitting modifications used here might suggest that the instrument has significant limitations and its usefulness is cast into doubt. Specifically, the use of the H-ISS as measures of activities or coping strategies for happiness that predict long-term improvements in well-being might not provide a way of distinguishing causes or consequences of emotional states, which are complex adaptive processes.

One way to improve the instrument could be to remove the items that have low loading on the latent factor or/and to remove the items that load on more that one factor. Yet another way of improving the instrument is to re-organize the items around a stronger theoretical basis rather than the factor analyses organization proposed by Tkach & Lyubomirsky (2006). For example, Cloninger (2004) has suggested that human thought can be organized across five planes based on the hierarchical development of the brain through evolution. Each of the planes corresponds to a specific modulation of a different basic emotional conflict: sexual, material, emotional, intellectual, and spiritual. We suggest that Cloninger’s five planes can be useful in a re-organization of the H-ISS and in the addition of new important strategies. Indeed, the H-ISS lacks items corresponding to basic needs, such as those corresponding to the sexual plane, and self-actualizing needs, such as those corresponding to the spiritual plane. Finally, we recommend that future research apply a qualitative method to get more deep information via open questions to participants. In this way participants are able to freely reflect upon complex adaptive processes.

Additional Information and Declarations

Competing Interests

Author Contributions

Human Ethics

Data Deposition

The authors declare there are no competing interests.

Ali Al Nima conceived and designed the experiments, performed the experiments, analyzed the data, contributed reagents/materials/analysis tools, wrote the paper, prepared figures and/or tables, reviewed drafts of the paper.

Danilo Garcia conceived and designed the experiments, performed the experiments, contributed reagents/materials/analysis tools, wrote the paper, reviewed drafts of the paper.

The following information was supplied relating to ethical approvals (i.e., approving body and any reference numbers):

After consulting with the Network for Empowerment and Well-Being’s Review Board we arrived at the conclusion that the design of the present study (e.g., all participants’ data were anonymous and will not be used for commercial or other non-scientific purposes) required only informed consent from the participants.

The following information was supplied regarding the deposition of related data:

Researchgate: https://www.researchgate.net/publication/277952371_RAW_DATA_-_Researchgate_Factor_Structure_of_the_Happiness-Increasing_Strategies_Scales_(H-ISS).

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
