# Peer review of "Factor structure of the happiness-increasing strategies scales (H-ISS): activities and coping strategies in relation to positive and negative affect"

_PeerJ, doi:10.7717/peerj.1059_

## Round 0.1 · original submission · Minor Revisions

Dear authors:

As you will see the reviewers believed this to be a good and publishable study. I agree with reviewer #1 that it would be greatly improved by minor revisions. I particularly feel that the Discussion needs rewriting: the reader should get a better idea of what you feel your results mean and why they are interesting and important. Please revise - you are close to the finish line!
Ada

Reviewer 1 ·

Basic reporting

The article is easily understood and well-described, even though there are several typographical errors and places where the English expression is not smooth. It needs to be carefully edited by a native English speaker or at least correct the following:
page 3, line 2, delete duplicated comma; page 6 lone 20, add "a" before Structural Equation Model; page 10, line 21, may belong should be "may be explained or attributed to"; line 22, whit should be with; line 24, delete "that"; page 11, line 8, make difference plural and gender plural; line 20, add the before present study; line 24, temporally should be temporary; page 12, line 2 rewrite as "significantly explain PA" and put parentheses around the r squared values; page 18, line 19 Table is misspelled.

Also I think the discussion is mostly a repetition of the results and could be more reflective. Is this really a useful inventory? It measures activities or coping strategies for happiness without really assessing whether this truly improves long-term satisfaction with life and doesn't provide a way of distinguishing causes or consequences of emotional states. I'm not taking a strong position about this, but there should be some discussion of how it can be used and what it does and doesn't do in a critical way. Most of the references are self-references, so perhaps this doesn't have wide interest.

Experimental design

The survey procedure is reasonable and well-described. The analytic procedure is appropriate and well-described. The model-fitting is typical of the problems encountered with the assumptions of SEM, which are perhaps unrealistic given that personality and emotionality are complex adaptive processes. As a result i large samples like this many correlations (here even to residuals) are added to force a fit. Perhaps there should be discussion of the limitations of linear models for this material.

Validity of the findings

The conclusions are well-justified overall. There is one variable loading that surprised me and that is not discussed. "Savor the moment" loads on "social affiliation" but it is hardly that. It is certainly possible to savor the moment in social isolation, so what does it mean that this is loading with the other behaviors involving social isolation? What does this tell us about being present, about adopting a savoring attitude, and its communality with social affiliation?

Additional comments

Overall the test has a reasonable factor structure and is descriptively useful. I'd like to know more about how the author thinks critically about its conceptual limitations and its utility.
Please copyedit the revision to remove typos and poor English grammar as suggested.

Please comment on the inadequacies of SEM as a description of dynamic human behavior -- it is ironic that an inventory of a coping strategy, which inherently involves reciprocal interactions to maintain homeostasis or happiness, is analyzed by a linear structural model. And of course your efforts to force a good fit were what people typically encounter when they use this method. You describe the problem but you do not comment on why it is hard to fit such linear models.

Reviewer 2 ·

Basic reporting

Good language. Much detail about the models are available for the interested reader.

Experimental design

Seems rigorous, well-planned and well carried out.

Validity of the findings

The authors discuss the literature regarding reliability of the use of MTurk. A further discussion of this could be of benefit to the paper.

Additional comments

Very interesting paper and the reviewer is impressed by the creative use of MTurk once again from this research group.

---

## Round 0.2 · accepted · Accept

Thank you for revising and resubmitting your manuscript. The current manuscript is accepted for publication. The discussion makes it clear now what your study contributes to our understanding of happiness increasing strategies. A weakness of the discussion is that you speak of prediction, while in fact modeling data that is taken at one time point. To the unsophistocated reader this might suggest causality, or at least precedence of one measurement over the other; however I believe the PeerJ readership will interpret this as meaning explained variance or association.